# Spin injection and inverse Edelstein effect in the surface states of topological Kondo insulator $SmB_6$

Qi Song[1,2], Jian Mi[1,2], Dan Zhao[3,4], Tang Su[1,2], Wei Yuan[1,2], Wenyu Xing[1,2], Yangyang Chen[1,2], Tianyu Wang[1,2], Tao Wu[3,4,5], Xian Hui Chen[3,4,5,6], X.C. Xie[1,2], Chi Zhang[1,2], Jing Shi[7] & Wei Han[1,2]

There has been considerable interest in exploiting the spin degrees of freedom of electrons for potential information storage and computing technologies. Topological insulators (TIs), a class of quantum materials, have special gapless edge/surface states, where the spin polarization of the Dirac fermions is locked to the momentum direction. This spin–momentum locking property gives rise to very interesting spin-dependent physical phenomena such as the Edelstein and inverse Edelstein effects. However, the spin injection in pure surface states of TI is very challenging because of the coexistence of the highly conducting bulk states. Here, we experimentally demonstrate the spin injection and observe the inverse Edelstein effect in the surface states of a topological Kondo insulator, $SmB_6$. At low temperatures when only surface carriers are present, a clear spin signal is observed. Furthermore, the magnetic field angle dependence of the spin signal is consistent with spin–momentum locking property of surface states of $SmB_6$.

[1] International Center for Quantum Materials, School of Physics, Peking University, Beijing 100871, China. [2] Collaborative Innovation Center of Quantum Matter, Beijing 100871, China. [3] Hefei National Laboratory for Physical Science at Microscale, Department of Physics, University of Science and Technology of China, Hefei, Anhui 230026, China. [4] Key Laboratory of Strongly-coupled Quantum Matter Physics, University of Science and Technology of China, Chinese Academy of Sciences, Hefei 230026, China. [5] Collaborative Innovation Center of Advanced Microstructures, Nanjing University, Nanjing 210093, China. [6] High Magnetic Field Laboratory, Chinese Academy of Sciences, Hefei, Anhui 230031, China. [7] Department of Physics and Astronomy, University of California, Riverside, California 92521, USA. Correspondence and requests for materials should be addressed to C.Z. (email: gwlzhangchi@pku.edu.cn) or to J.S. (email: jing.shi@ucr.edu) or to W.H. (email: weihan@pku.edu.cn).

Spintronics aims to use the spin degrees of freedom for information technologies[1-3]. The injection of spin-polarized carriers into two-dimensional quantum materials, including graphene and the surface states of topological insulators (TIs), is particularly interesting[4,5]. Different from graphene showing weak spin–orbit coupling and long spin lifetimes[6-9], the surface states of TI exhibit very large spin–orbit coupling[10-13]. Even more interestingly, the spin and the momentum directions are strongly coupled to each other in the surface states of TI[4,10-14]. Since the observation of the spin–momentum locking properties with scanning tunneling microscopy and spin-angle-resolved photoemission spectroscopy (spin-ARPES)[15,16], a great deal of effort has been made to demonstrate various unique effects associated with this property, such as large spin polarization currents and large spin–orbit torque in the $Bi_2Se_3$-based three-dimensional TI[17-25]. However, a major obstacle to the clean demonstration of the Edelstein/inverse Edelstein effects for the spin–momentum locked surface states is the presence of unavoidable bulk carriers which dominate the conduction in these $Bi_2Se_3$-based three-dimensional TI[19,26]. Recently, $SmB_6$, a Kondo insulator, has been found to be a new type of TI based on transport measurements and ARPES[27-35]. At temperatures below $\sim 3\,K$, the bulk states are insulating, and only surface carriers contribute to the conduction, as demonstrated by the previous surface Hall measurements[30,31].

Here, we report the spin injection into the surface states using the spin pumping and the observation of the inverse Edelstein effect in this topological Kondo insulator (TKI). The temperature and magnetic field angle dependences of the spin voltage are consistent with the spin–momentum locking properties of the surface states, which have been shown to be topological in previous studies[29].

## Results

**Spin injection into the surface states of $SmB_6$.** The spin injection experiment is performed using $Ni_{80}Fe_{20}$ (Py) as the spin injector, which is deposited onto the (001) surface of the $SmB_6$ single crystals, as shown in Fig. 1a (see the 'Methods' for details). When the ferromagnetic resonance condition for Py is fulfilled under certain magnetic fields and microwave frequencies, the precessing magnetization launches a spin current, which enters the adjacent nonmagnetic $SmB_6$ layer. This technique is called spin pumping, which has been widely used to measure the spin to charge conversion in various materials, including metals, semiconductors and graphene and so on[36-43]. In our measurements, we use a radio frequency (RF) signal generator to provide the microwave power and standard lock-in technique for better sensitivity and signal-to-noise ratio (see the 'Methods' for details). Figure 1b shows the schematic drawing of energy dispersion relationship of the surface states at the Fermi level for both $\bar{X}$ and $\Gamma$ points. The resistance of the $SmB_6$ device is measured from 300 to $\sim 0.8\,K$, as shown in Fig. 1c. Clearly, the resistance saturates blow $\sim 3\,K$, which indicates that the surface states are dominant and the bulk states do not contribute to conduction. As the temperature increases, the resistance decreases quite rapidly, owing to a large number of the activated bulk carriers as the temperature increases.

Figure 1d shows the typical magnetic field dependence of the spin voltage measured at $1.7\,K$ with three representative microwave frequencies of 8.3, 9.4 and 10.1 GHz, respectively. We first confirm that the magnetic fields, at which we observe the voltage signals, are the same as the resonance magnetic fields ($H_{res}$) of the Py under the same microwave frequencies (Supplementary Fig. 1 and Supplementary Note 1). It is noticed that there are mainly three contributions to the voltages, namely

the voltage due to the spin pumping and inverse Edelstein effect ($V_{SP}$), the voltage due to the Seebeck effect from the microwave heating ($V_{SE}$) and the anomalous Hall effect ($V_{AHE}$) of the Py. Due to their different symmetries as a function of the magnetic field, we can obtain the voltage amplitudes of all these three contributions by fitting the magnetic field dependence of the voltage with the following equation (Supplementary Fig. 2 and Supplementary Note 2).

$$V(H) = V_S \frac{(\Delta H)^2}{(H-H_{res})^2 + (\Delta H)^2} + V_{AS} \frac{-2\Delta H(H-H_{res})}{(H-H_{res})^2 + (\Delta H)^2} \quad (1)$$

where $V_S$ and $V_{AS}$ are the voltage amplitudes for the symmetric and antisymmetric Lorentzian shapes, respectively, and $\Delta H$ is the half-line width. The $V_{SP}$ exhibits a positive sign for positive magnetic fields and the positive sign of the spin-to-charge conversion in the surface states of the $SmB_6$ is theoretically expected from the counter-clockwise spin textures for the electron band of the topological surface states[18,42]. The counter-clockwise spin textures have been shown by both spin-ARPES measurements and DFT calculations[34,44]. After the determination of $H_{res}$ and $\Delta H$ for all applied microwave frequencies, we obtain the effective magnetization ($M_{eff}$) and the Gilbert damping constant for the Py layer. Our results show that $M_{eff}$ is $781 \pm 16$ e.m.u. $cm^{-3}$, which is obtained using the Kittel formula shown below[45]:

$$f_{res} = \left(\frac{\gamma}{2\pi}\right)[H_{res}(H_{res} + 4\pi M_{eff})]^{1/2} \quad (2)$$

where $\gamma$ is the geomagnetic ratio. From the slope of the linearly fitted curve of the half-line width versus microwave frequency at $1.7\,K$, we calculate the Gilbert damping constant of the Py on $SmB_6$ to be $0.0166 \pm 0.0006$ (Supplementary Fig. 3).

The microwave power dependence of the spin voltage is shown in Fig. 2a measured at $1.7\,K$ and with the microwave frequency of 10.1 GHz. The measured resonance peak increases as the microwave power increases. Following the same fitting procedure (Supplementary Note 2), we obtain the power dependence of $V_{SP}$ and $V_{SE}$. Both $V_{SP}$ and $V_{SE}$ show a linear relationship with the microwave power, as shown in Fig. 2b,c.

**Temperature dependence of the spin voltage.** As mentioned earlier, the surface states of $SmB_6$ dominate the transport as the bulk carriers freeze out below $\sim 3\,K$; above $\sim 3\,K$, the contribution from the bulk states is thermally activated. When a spin current enters the spin–momentum locked surface states, an electric field is resulted due to the inverse Edelstein effect, which is measured as a spin voltage. To investigate how the spin voltage evolves as the surface states emerge and become dominant, we perform the measurements from $\sim 0.8$ to 10 K. Below $\sim 0.8\,K$, it is difficult to stabilize the temperature due to the microwave heating. Figure 3a shows the typical measurements of the voltage as a function of the magnetic field with the microwave power of 100 mW and frequency of 10.1 GHz at 0.84, 1.66, 2.1, 2.3 and 10 K, respectively. At $0.8\,K$, when only spin–momentum locked surface states exist, the spin signal is $\sim 42$ nV. This value is relatively small compared with previous studies on $Bi_{1.5}Sb_{0.5}Te_{1.7}Se_{1.3}$ and $\alpha$-Sn (refs 18,42), which could be related to the spin pumping efficiency and/or the spin-to-charge conversion efficiency and needs further studies (Supplementary Note 3). The spin voltage steadily decreases as the temperature increases, and when the temperature reaches 10 K, no voltage can be detected. The resistance of the $SmB_6$ from 10 to $\sim 0.8\,K$ is shown in Fig. 3b, indicating that the bulk states start to contribute to the total conductance between 2 and 3 K. From 3 to 5 K, the conduction due to the bulk carriers quickly increases, resulting in

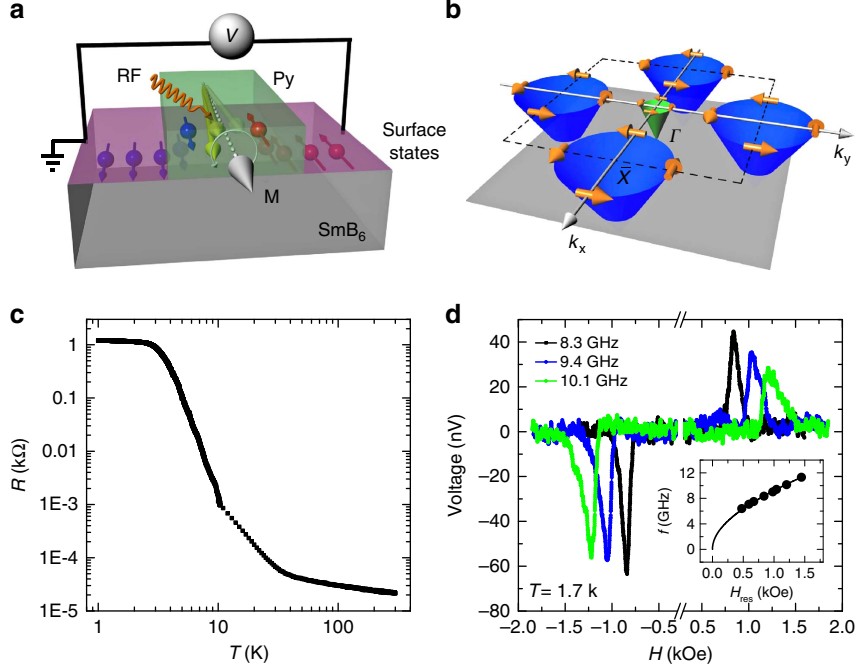

**Figure 1 | Spin injection into the surface states of SmB$_6$.** (**a**) Schematic drawing of device structure and the spin injection and inverse Edelstein effect measurements. (**b**) Schematic drawing of the spin–momentum locking properties of the topological surface states at the $\bar{X}$ and $\Gamma$ points based on previous photoemission spectroscopy measurements and DFT calculations[34,44]. (**c**) The resistance of the SmB$_6$ as a function of the temperature. (**d**) Typical magnetic field dependence of the voltage with various GHz microwave frequencies. The power of the microwave is 100 mW and the temperature is 1.7 K. Inset: the resonance frequency ($f$) as a function of the resonance magnetic field ($H_{res}$). The solid line is a fitted curve based on the Kittel formula, equation (2) in the main text.

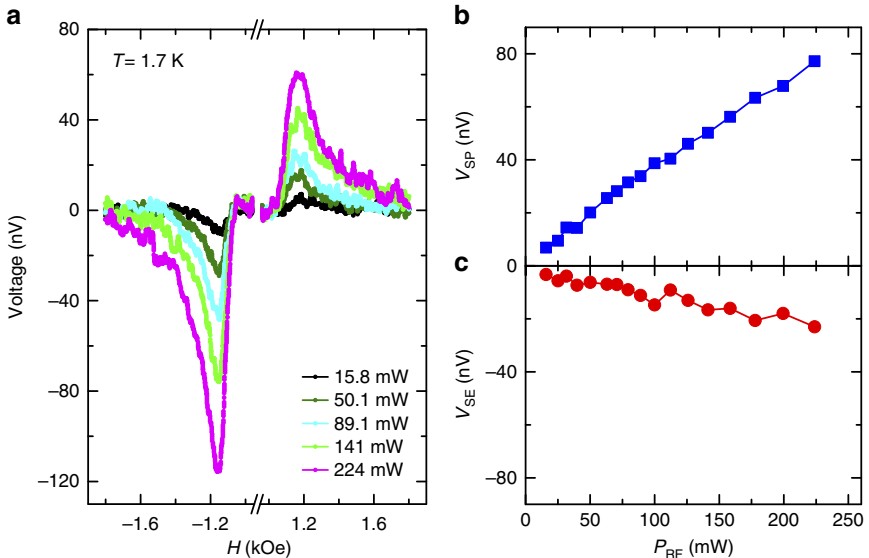

**Figure 2 | Microwave power dependence of the spin injection into the surface states of SmB$_6$.** (**a**) Magnetic field dependence of the voltage measured at the temperature of 1.7 K and with the microwave frequency of 10.1 GHz and power of 15.8, 50.1, 89.1, 141 and 224 mW, respectively. (**b,c**) Microwave power dependence of the measured voltage due to spin pumping and inverse Edelstein effect ($V_{SP}$ in **b**) and the voltage that is related to the Seebeck effect ($V_{SE}$ in **c**).

a 100-fold decrease in the total resistance. This feature is consistent with the previous surface conductance and Hall measurements, indicating the nearly pure surface states contributing to the conduction[30,31]. The temperature dependence of the $V_{SP}$ is summarized in Fig. 3c. $V_{SP}$ shows little temperature dependence below ~2.2 K. At temperatures above ~2.2 K, $V_{SP}$ steadily decreases as the temperature increases. The temperature dependences of both $V_{SP}$ and the resistance strongly support

that the spin signal originates from the spin–momentum locked surface states. When the spin polarization is generated in the surface states, an in-plane electrical voltage is produced in the direction perpendicular to the spin directions, due to the inverse Edelstein effect. As the temperature further increases, more bulk carriers are activated and the spin voltage is greatly suppressed. This is very interesting, for the bulk states should have strong spin–orbit coupling as well and therefore ordinary inverse spin

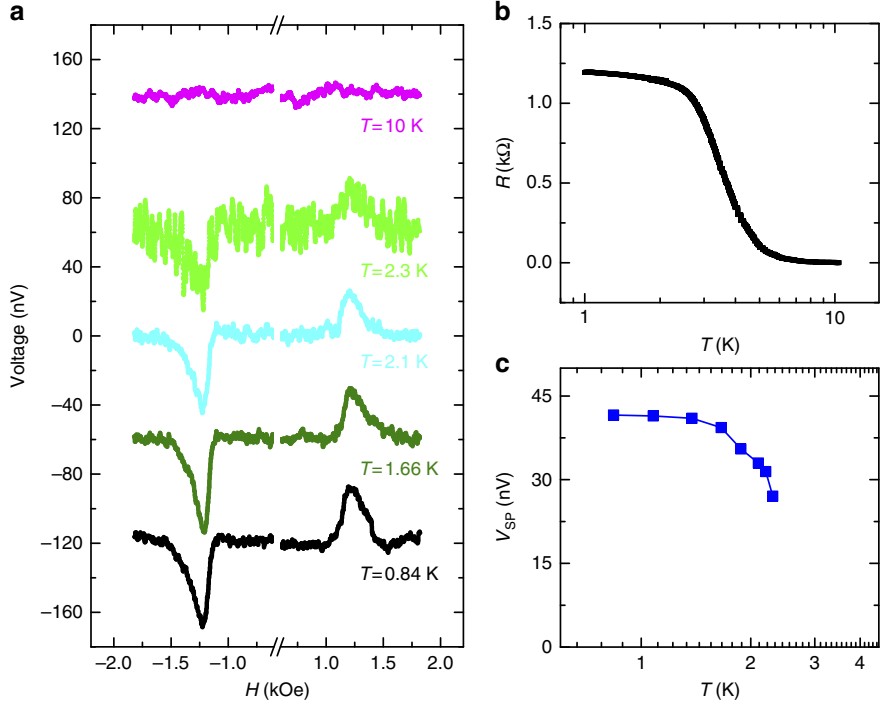

**Figure 3 | Temperature dependence of the spin injection into the surface states of SmB₆.** (**a**) Magnetic field dependence of the voltage measured for the temperatures of 0.84, 1.66, 2.1, 2.3 and 10 K, respectively. The measurement is performed with a microwave power 100 mW and frequency of 10.1 GHz. (**b**) The resistance of the SmB₆ as a function of the temperature from 10 to ∼0.8 K. (**c**) Temperature dependence of $V_{SP}$.

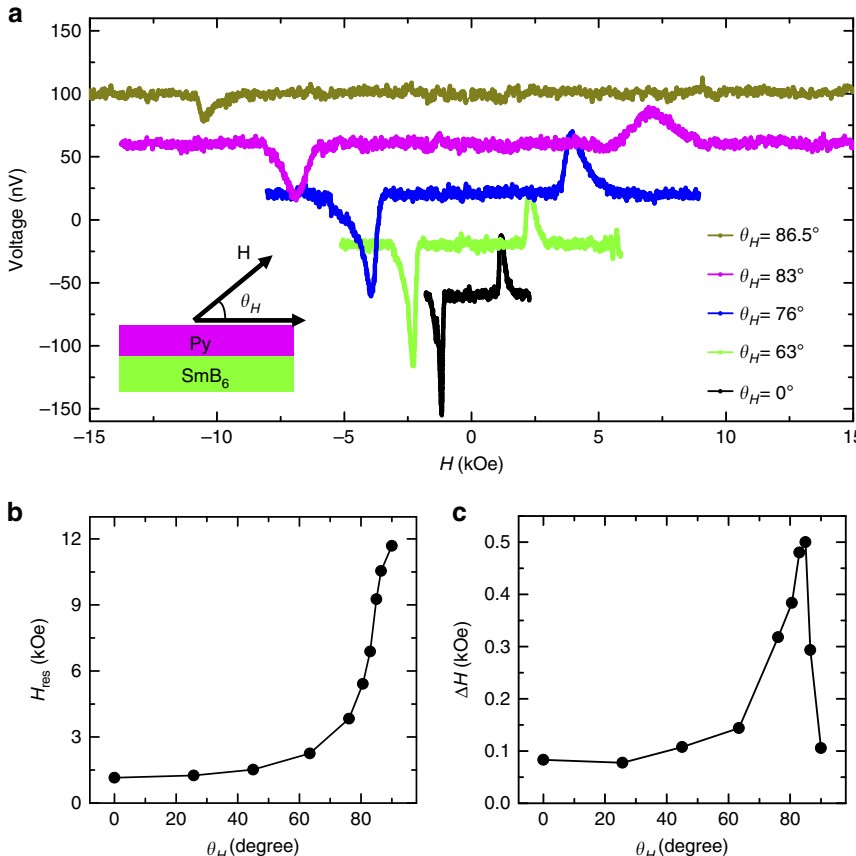

**Figure 4 | The measured voltage as a function of the magnetic field angle.** (**a**) Magnetic field dependence of the voltage measured at 1.7 K for $\theta_H = 0°$, 63°, 76°, 83° and 86.5°, respectively. Inset: the schematic illustration of the coordinate system for magnetic field angle. (**b,c**) The resonance magnetic field and half-line width as a function of $\theta_H$.

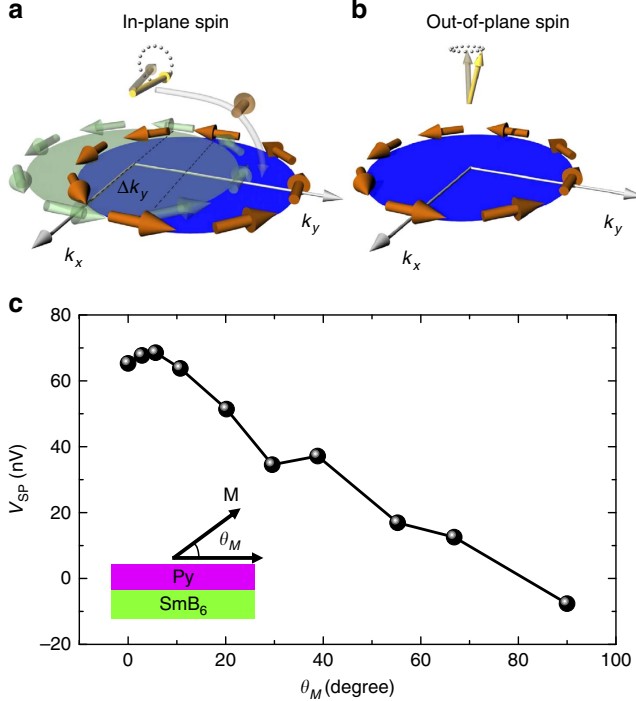

**Figure 5 | Magnetization angle dependence of the voltage due to spin pumping and inverse Edelstein effect.** (**a**,**b**) Schematic drawings for the in-plane (**a**) and out-of-plane (**b**) spin polarization injection into the surface states of $SmB_6$. The in-plane spin polarization injection leads to the generation of the in-plane electric field due to inverse Edelstein effect, while the out-of-plane spin injection is forbidden due to the spin–momentum locking properties of the surface states of $SmB_6$, which have been shown to be topological in previous studies. (**c**) $V_{SP}$ as a function of $\theta_M$. Inset: the schematic illustration of the coordinate system for $\theta_M$.

Hall effect from bulk states could give rise to a finite voltage. However, we do not observe any voltage signal at high temperatures.

**Magnetic field angle dependence of the spin voltage.** To further confirm the spin injection and detection in the surface states of the TKI, $SmB_6$, we study the in-plane and out-of-plane spin polarization injection by changing the magnetic field direction. Figure 4a shows the typical results of the magnetic field-dependent voltages at 1.7 K with a microwave power of 200 mW and frequency of 10.1 GHz for the angles between the magnetic field and the Py electrode (shown in the inset figure), $\theta_H$, equal to 0°, 63°, 76°, 83° and 86.5°. As $\theta_H$ increases, the resonance magnetic field increases accordingly, and in the meantime, the spin signal shows a decrease. At 86.5°, the spin-dependent voltage becomes vanishingly small. The $H_{res}$ and $\Delta H$ as a function of $\theta_H$ are shown in Fig. 4b,c, which are consistent with the previous measurement of the ferromagnetic resonance of Py under different magnetic field directions[37,46]. This further confirms that the measured spin voltage indeed arises from the precession of the Py magnetization.

## Discussion

It is particularly interesting that only in-plane spin polarization injection generates an electric field, whereas the out-of-plane spin polarization injection does not show this effect. This observation could be attributed to the spin–momentum locking properties of the surface states of the TKI, as illustrated in Fig. 5a,b. For the in-plane spin polarization injection along the $x$ direction, the

Fermi surface shifts along the y direction, and $\Delta k_y$ indicates the total shift due to the spin injection and the inverse Edelstein effect, as shown in Fig. 5a. On the other hand, for the out-of-plane spin polarization injection, there is no net effect of spin injection as the spins of the surface states lie in-plane and are locked perpendicular to the momentum directions, as shown in Fig. 5b. Finally, we calculate the Py magnetization angle, $\theta_M$, from the $\theta_H$ dependence of the resonance magnetic field (Supplementary Fig. 4) based on the 0 and 90 degrees data and the following equation[37].

$$2H_{res}\sin(\theta_H - \theta_M) - 4\pi M_S \sin(2\theta_M) = 0 \qquad (3)$$

where $M_S$ is the saturated magnetization. It is clearly seen that $V_{SP}$ almost vanishes as $\theta_M$ approaches 90 degrees (Fig. 5c), which is also consistent with the spin–momentum locking properties of the surface states of the TKI, as discussed above and illustrated in Fig. 5a,b. The complete understanding of the $V_{SP}$ as a function of the $\theta_M$ needs future theoretical studies to quantitatively calculate how much the Fermi surface shift as a result of the inverse Edelstein effect of the spin polarization injection (Supplementary Fig. 5 and Note 4).

Our experimental results strongly support the demonstration of spin injection and the observation of the inverse Edelstein effect in the surface states of $SmB_6$. The temperature and magnetization angle dependences, as well as the sign of the spin-to-charge conversion are well consistent with spin–momentum locking properties of the surface states, which have been shown to be topological with the counter-clockwise spin textures for the electron bands in previous studies[29,34,44]. Since the detailed spin textures of the Rashba surface states have not been reported yet, it is premature to exclude any contribution from the Rashba-split surface states at the current stage. To fully understand this, further studies, including the detailed spin textures from spin-ARPES measurements of the Rashba surface states and the quantitative theoretical calculations of the contributions from topological and Rashba surface states, are needed. Our observation could lead to future studies of the role of strong correlation in TKIs for spintronics and highly efficient spin current generation in the surface states of TIs via the materials design and engineering.

## Methods

**Materials growth.** High-quality single crystalline $SmB_6$ samples are grown using the conventional Al-flux method. A mixture consisting of a Sm chunk (purity: 99.9%), Boron (purity: 99.99%) and Al powders (purity: 99.99%) with a ratio of 1:6:400 is heated at high temperatures in the circumstance with continuously flowing Ar gas to form $SmB_6$ single crystals. Then the $SmB_6$ samples are put into diluted $HNO_3$ acid to remove the residual aluminum flux.

We choose the samples with large rectangular crystals of millimeters size and large (001) facet for spin injection experiment. A 20 nm thick Py electrode is deposited on the (001) surface of the $SmB_6$ single crystal by radio frequency magnetron sputtering with a growth rate of 0.02 Å s$^{-1}$. To prevent the oxidation of Py, a capping layer of 3 nm Al is deposited *in situ* before taking the samples out.

**Device fabrication.** A shadow mask technique (size: ~0.9 × 3 mm$^2$) is used to define the shape and position of the ferromagnetic electrode (Py/Al) on the (001) surface of the $SmB_6$ crystal (size: ~1 × 5 mm$^2$, thickness: ~0.5 mm). Al wires are used to contact the two ends of $SmB_6$ sample for the electrical voltage measurement.

**Device measurement.** The spin injection is performed using the spin pumping method and the spins are detected via the inverse Edelstein effect of the surface states of $SmB_6$. The microwave power is supplied by a signal generator (Anritsu LTD. MG3690C) modulated with a digital lock-in amplifier (NF Co. LI5640) with the frequency of 373 Hz to enhance the sensitivity and signal-to-noise ratio. The spin pumping measurement is performed by precessing the Py magnetization around its resonance conditions with a coplanar waveguide from 10 to ~0.8 K in a Janis He-3 system. The resistance of the $SmB_6$ single crystal is measured using Keithley K2400 and K2002 in Quantum Design Physical Properties Measurement System (PPMS) from 300 to 10 K and in a Janis He-3 system from 10 to ~0.8 K.

**Data availability.** The authors declare that the data supporting the findings of this study are available within the paper and its Supplementary Information files.

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

## Acknowledgements

We acknowledge the fruitful discussions with Professor Fa Wang and the financial support from National Basic Research Programs of China (973 program Grant Nos. 2014CB920902, 2015CB921104 and 2013CB921903) and National Natural Science Foundation of China (NSFC Grant Nos. 11574006 and 11374020). D.Z., T.W. and X.H.C. acknowledge the financial support from the Strategic Priority Research Program of the Chinese Academy of Sciences (Grant No. XDB04040100). T.W. acknowledges Recruitment Program of Global Experts and CAS Hundred Talent Program. J.S. acknowledges the support by the DOE BES Award No. DEFG02-07ER46351. W.H. acknowledges the support by the 1000 Talents Program for Young Scientists of China.

## Author contributions

W.H. proposed and designed the experiment. Q.S. did the Py growth and fabricated the devices. Q.S. and J.M. performed the electrical measurements. J.M. and C.Z. developed the microwave techniques in the He3 refrigerator used for the measurements below 10 K in Professor Zhang's group. Q.S. and W.H. analysed the data. D.Z., T.W. and X.H.C. provided the single crystalline $SmB_6$ sample. W.H. wrote the manuscript. All the authors commented on the manuscript and contributed to its final version.

## Additional information

**Competing financial interests:** The authors declare no competing financial interests.

