## [Peer Review File · Nature Communications]

Reviewers' comments:

Reviewer #1 (Remarks to the Author):

When the current is driven through the two-dimensional electron gas with large spin-orbit coupling, it induces a spin-polarization in perpendicular direction to the current. This effect is known as Edelstein effect. This manuscript reports on the observation of the inverse Edelstein effect as well as detection of the of the electronic spin-texture on the surface of samarium hexaboride (SmB₆) via spin pumping. In particular, the authors claim that they detect the topological surface states in a Kondo insulator, SmB₆.

While I do not question validity and the results of the experiment, I am concerned with the interpretation of the data. I am sure that the authors are well aware of the existing controversy concerning the nature and microscopic structure of the surface states in SmB₆. In particular, apart from the "topological surface states" scenario, several groups proposed that the surface states in SmB₆ are topologically trivial and, as a consequence, the electrons have parabolic dispersion, but due to the spin-orbit coupling on the surface there are two or more "Rashba-split" notes. Unfortunately, the present experiment cannot distinguish between linear vs. quadratic dispersion relation for the surface electrons.

This is why I am not convinced that the authors have indeed detected topological surface states as they claimed in the abstract. Therefore, I believe that this manuscript is not suitable for the publication in Nature Communications.

Reviewer #2 (Remarks to the Author):

The manuscript "Spin Injection and Detection in the Surface States of a Kondo Topological Insulator SmB₆" by Song et al. studied the temperature and magnetic field angle dependence of the spin signal due to spin pumping in Kondo topological insulator SmB₆. The topological insulators could enable efficient spin detection, the research topic in the current study is very important and timely for spintronics applications. These dependence of the spin signal in SmB₆/ferromagnetic material hetero-structure have not been studied before. For the reason, this work may deserve for publication in Nature Communications. However, the interpretation of their experimental results could be serious flawed, as I pointed out in below.

1. The expression of "Spin injection and detection" in title may be lead to misunderstanding as "spin injection and spin detection".
2. There are many reports about proximity effect in TI/ferromagnetic material [P. Wei et al., PRL 110, 186807 (2013).etc...]. Is there the influence of the effect in your experimental result?
3. Which sign of spin-to-charge conversion? Is the sign consistent with that of ARPES results(ref.34)?
4. In line 122 (page 6) of your main text, there is interpretation of temperature dependence, "However, we do not observe any voltage signal at high temperatures. This feature indicates that the inverse Edelstein mechanism offers highly efficient spin-charge conversion due to the spin-momentum locking property of the TI surface states."

However there is the leap in the interpretation. In generally, the generated voltage due to spin-to-charge conversion decrease with decreasing the sample resistance. When sample resistance at 10K becomes to one-hundredth of that of low temperature, the generated voltage should be one-hundredth. Maybe it is difficult to detect the voltage even though the conversion efficiency at 10K is the same with that of low temperature. So these results are not clear evidence of efficient spin-charge conversion. If author want to claim the enhancement of spin-charge conversion at low temperature,

author should show the temperature dependence of conversion efficiency. And why the noise at $T=2.3\text{K}$ is large?

5. Figure 5 shows magnetization angle dependence of the voltage due to spin pumping. The magnetic field angle dependence is the same with conventional spin Hall effect [Ando et al, PRB 78 014413 (2008)]. It is difficult to claim that the voltage is caused by Edelstein effect from the magnetic field angle dependence.

Reviewer #3 (Remarks to the Author):

This manuscript reports on experimental demonstration of the spin to charge conversion via the surface states of the Kondo insulator SmB₆. Generated by the spin pumping originating from the FMR of Py, the spin current is injected into the surface states of SmB₆, yielding the spin to charge conversion due to the inverse Edelstein effect (IEE) arising from the spin texture of the surface states. The realization of IEE is clearly confirmed by the field angle and temperature dependence of the observed spin signal.

This work is intriguing, the manuscript is well-written, and scientifically solid data clearly support the observation of spin injection and detection in the surface states of the strongly correlated material for the first time. However, the interpretation of the origin of IEE, namely, the spin-momentum locking, appears not to be convincing because IEE could occur in two dimensional electron systems with strong Rashba splitting in the band structures which could be realized in SmB₆ because of its highly reactive surface. Therefore, I would recommend this paper for publication in Nature Communications after some revisions.

In addition, it seems that there're no figures included in the supplementary information. Please compile the revised manuscript correctly.

My remarks to the manuscript:

1. The authors claim that the observed spin signal stems from the IEE associated with the spin-momentum locking of the topologically non-trivial surface states in SmB₆. However, the IEE can be observed in topologically "trivial" two dimensional states induced by strong Rashba splitting in the band structure. (For instance, see J.C. Sanchez et al., Nat. Commun. 4 2944 (2014).)

Although extensive experimental studies support the presence of the surface states in SmB₆, the topologically non-trivial aspects have been confirmed only by a few reports, including the torque magnetometry (G. Li et al. Science 346, 1208 (2014)), spin resolved ARPES (N. Xu et al., Nat. Commun. 5, 4566 (2014)), and one-dimensional chiral magnetotransport (Y. Nakajima et al. Nat. Phys. 12, 213 (2016)). Besides, even polarity-driven topologically-trivial surface states has been observed by ARPES measurements (Z. Zhu, et al., Phys. Rev. Lett. 111, 216402 (2013)). It seems to me that the surface of SmB₆ is quite reactive, variously terminated and can reconstruct, providing different things observed by various groups. In this situation, it is unclear whether the spin-momentum locked surface states in SmB₆ give rise to the IEE.

However, it would be instructive to know if the observed spin signal quantitatively exclude the presence of topologically "trivial" surface states in SmB₆. Otherwise, the authors should use moderate wording to describe the origin of the spin signal.

2. As mention above, the Fermi surfaces observed by ARPES measurements seem to be inconsistent with each other. What literature is the schematic drawing in Fig.1b based on? Experiment or theory? The authors should cite the papers on which the drawing is based.

3. It would be instructive to compare the value of the observed spin signal with that observed in the spin injection into weakly correlated TIs. For instance, similar spin-charge conversion in BSTS/Ni₈₁Fe₁₉ device has been reported (Y. Shiomi et al., PRL 113, 196601 (2014)), showing much larger spin signal (~10 micro volts) induced by much smaller microwave power than the present work. This fact appears to evince that the IEE in SmB₆ might be induced by trivial Rashba splitting surface states because one of the pair of splitting bands counteracts the effect of the other. The authors should elaborate on the difference between the signals observed in SmB₆ and weakly correlated TI devices.

Also, the observed sign of the spin signal in SmB₆ is opposite to that in the BSTS/Ni₈₁Fe₁₉ device. I'm curious to know if it's due to intrinsic mechanism or just the difference of notation.

4. The presence of the bulk carriers is a major obstacle to see the EE/IEE in (weakly correlated) TIs. Truly bulk-insulating at low temperatures, Kondo insulator SmB₆ is one of the suitable materials for the spin injection and detection through the IEE. Again, to compare SmB₆ with other TIs, it would be very helpful to quantify the spin-injection efficiency, which can be associated with leakage into bulk carriers, if possible.

5. It is known that a native surface oxide layer is observed by X-ray photoelectron spectroscopy for SmB₆ (W.A. Phelan et al., Phys. Rev. X 4, 031012 (2014)), which could affect the efficiency of spin injection. How did the author treat the surface of SmB₆ while fabricating the device?

6. It would be instructive to know the typical length scales in SmB₆, such as the sample thickness, the spin diffusion length in the surface state, and the penetration depth of the surface states, to figure out the properties of the device in the present work.

REVIEWERS' COMMENTS:

Reviewer #1 (Remarks to the Author):

I very much appreciate the efforts of the authors to address the concerns I have raised in my report. Nevertheless, based on the author's response I believe that the current work only adds to the controversy surrounding SmB6 on the possible topological nature of the surface states as opposed to resolving it. Therefore, I do not recommend this paper for the publication in Nature Communications.

Reviewer #2 (Remarks to the Author):

Authors have addressed the points raised in the previous review with varying degrees of satisfaction. I believe that the detected signals are caused by spin-to-charge conversion in surface state of topological Kondo Insulator SmB6. So I would be happy to see the manuscript accepted for publication at Nature Communications.

However, I could not understand whether the amplitude of detected signal is reasonable compared with ideal case. So authors should compare the amplitude of generated voltage (or conversion efficiency) with previous works in several topological insulators like alpha-Sn (Phys. Rev. Lett. 116, 096602). If you found that detected signal is tiny, you should add the comments about the result in main text.

Reviewer #3 (Remarks to the Author):

The authors have provided a reasonable and satisfactory reply to my comments, and the manuscript has been improved. Therefore, the paper meets all the standards for publication in Nature Communications, and I recommend publication as it is.

Detailed responses to Referees

We address the minor issues raised by the referees below (referees' comments in blue and italics and our responses in black).

Response to Referee #1:

- 1) *This manuscript reports on the observation of the inverse Edelstein effect as well as detection of the of the electronic spin-texture on the surface of samarium hexaboride (SmB₆) via spin pumping. In particular, the authors claim that they detect the topological surface states in a Kondo insulator, SmB₆. I am concerned with the interpretation of the data. I am sure that the authors are well aware of the existing controversy concerning the nature and microscopic structure of the surface states in SmB₆. In particular, apart from the "topological surface states" scenario, several groups proposed that the surface states in SmB₆ are topologically trivial and, as a consequence, the electrons have parabolic dispersion, but due to the spin-orbit coupling on the surface there are two or more "Rashba-split" notes. Unfortunately, the present experiment cannot distinguish between linear vs. quadratic dispersion relation for the surface electrons. This is why I am not convinced that the authors have indeed detected topological surface states as they claimed in the abstract.*

Reply: We thank the referee for pointing out this important issue.

Firstly, we apologize for using the misleading term of “*spin injection and detection*” in the title and some other places in our previous manuscript. We meant to use this term to describe the spin pumping into the surface states and the detection of the injected spins via the inverse-Edelstein effect. We did not aim to claim that we were able to detect the spin texture of the surface states of SmB₆, due to the limitation of our measurement technique. To make this clear, we have made the following two changes.

- 1) We revised the title to be “*Spin Injection and Inverse Edelstein Effect in the Surface States of Topological Kondo Insulator SmB₆*”.
- 2) In the abstract, we have changed “*demonstrate the spin injection and detection*” to “*demonstrate the spin injection and observe the inverse Edelstein effect*”.

Secondly, the major contribution of our results to this field is that we have clearly demonstrated the spin injection and observed the inverse Edelstein effect of the surface states of topological Kondo insulator SmB₆. And the most straightforward explanation is the spin-momentum locking properties of the surface states that have been proven to be topological by most studies in this field (See the recent review paper: Annu. Rev. Condens. Matter Phys. 7, 249-280, (2016) and the reference in there). Furthermore, we have carefully studied the sign of the spin-to-charge conversion and found the sign is consistent with spin-momentum locking properties of the topological surface states with the counter-clockwise spin textures for the

electron bands, which have been supported by recent spin-APERS measurements and DFT calculations (Xu et al, Nat. Commun. 5, (2014) and Yu et al, arXiv:1603.09677 (2016)).

Thirdly, we used the SmB₆ compounds (provided by Prof. Xianhui Chen's group) which have been proved to exhibit topological surface states from both quantum oscillation and APERS measurements (Li et al, Science 346, 1208-1212, (2014) and Jiang et al, Nat. Commun. 4, (2013)). Thus, we are very confident that our SmB₆ samples indeed host topological surface states. There are some other groups reporting the non-topological surface states on the SmB₆, which might be related to different synthesis methods. However, the difference is beyond the studies in this manuscript.

Fourthly, we agree that we cannot fully exclude contribution to the spin-to-charge conversion from the possible "Rashba-split states". However, since the detailed spin textures of the Rashba surface states has not been reported yet, it is premature to exclude origin of the spin-momentum locking from the Rashba surface states at the current stage. To fully understand this, further studies, including the detailed spin textures from spin-APERS measurements of the Rashba surface states and the quantitative theoretical calculation of the contributions from topological and Rashba surface states, are needed.

We have also made the following revisions to the manuscript:

- 1) We added the following sentences on page 4, line 17: *“The V_{SP} exhibits a positive sign for positive magnetic field and the sign of the spin-to-charge conversion in the surface states of the SmB₆ is theoretically expected from the counter-clockwise spin textures for the electron band of the topological surface states [Shiomi, et al, PRL 113, 196601 (2014) and Rojas-Sánchez, et al, PRL, 116, 096602 (2016)]. The counter-clockwise spin textures have been both shown by spin-APERS measurements and DFT calculations [Xu et al, Nat. Commun. 5: 4566(2014) and Yu et al, arXiv:1603.09677 (2016)].”*
- 2) We revised the summary section to be: *“Our experimental results strongly support the demonstration of spin injection and the observation of the inverse Edelstein effect in the surface states of SmB₆. The temperature and magnetization angle dependence, as well as the sign of the spin-to-charge conversion are well consistent with spin-momentum locking properties of the surface states, which have been shown to be topological with the counter-clockwise spin textures for the electron bands in previous studies [Xu et al, Nat. Commun. 5, (2014) and Yu et al, arXiv:1603.09677 (2016)]. Since the detailed spin textures of the Rashba surface states have not been reported yet, it is premature to exclude any contribution from the Rashba splitted surface states at the current stage. To fully understand this, further studies, including the detailed spin textures from spin-APERS measurements of the Rashba surface states and the quantitative theoretical calculation of the contributions from topological and Rashba surface states, are needed.”*

Overall, believe that we have clearly demonstrated the spin injection and observed the inverse

Edelstein effect of the surface states of SmB_6 , which exhibit spin-momentum locking properties. And the most straightforward explanation is the spin-momentum locking properties of the surface states that have been shown to be topological in most studies done recently.

Response to Referee #2:

1) *The expression of "Spin injection and detection" in title may be lead to misunderstanding as "spin injection and spin detection".*

Reply: We thank the referee for pointing out this misunderstanding. We first revised the title to be "*Spin Injection and Inverse Edelstein Effect in the Surface States of a Topological Kondo Insulator SmB_6* ". We also replaced the words of "detection" by "*observation of the inverse Edelstein effect*" for the whole paper.

2) *There are many reports about proximity effect in TI/ferromagnetic material [P. Wei et al., PRL 110, 186807 (2013).etc...]. Is there the influence of the effect in your experimental result?*

Reply: We thank the referee for pointing out this issue. We are not sure whether there is proximity effect between SmB_6 and Py, which requires future studies. If there is proximity effect, the spin pumping efficiency and the spin mixing conductance could be affected.

3) *Which sign of spin-to-charge conversion? Is the sign consistent with that of ARPES results (ref.34)?*

Reply: We thank the referee for pointing out this important issue. Yes, it is consistent with the counter-clockwise spin textures for the electron band of the topological surface states. We added the following sentences on page 4, line 17: "*The V_{SP} exhibits a positive sign for positive magnetic field and the sign of the spin-to-charge conversion in the surface states of the SmB_6 is theoretically expected from the counter-clockwise spin textures for the electron band of the topological surface states [Shiomi, et al, PRL 113, 196601 (2014) and Rojas-Sánchez, et al, PRL, 116, 096602 (2016)]. The counter-clockwise spin textures have been both shown by spin-APERS measurements and DFT calculations [Xu et al, Nat. Commun. 5: 4566(2014) and Yu et al, arXiv:1603.09677 (2016)].*"

4) *In line 122 (page 6) of your main text, there is interpretation of temperature dependence, "However, we do not observe any voltage signal at high temperatures. This feature indicates that the inverse Edelstein mechanism offers highly efficient spin-charge conversion due to the spin-momentum locking property of the TI surface states.". However there is the leap in the interpretation. In generally, the generated voltage due to spin-to-charge conversion decrease with decreasing the sample resistance. When sample resistance at 10K becomes to one-hundredth of that of low temperature, the generated voltage should be one-hundredth. Maybe it is difficult to detect the voltage even though the conversion efficiency at 10K is the*

same with that of low temperature. So these results are not clear evidence of efficient spin-charge conversion. If author want to claim the enhancement of spin-charge conversion at low temperature, author should show the temperature dependence of conversion efficiency. And why the noise at T=2.3K is large?

Reply: We thank the referee for pointing out this issue. We agree with the referee that the current shunting could also be one explanation. To make it clear, we have deleted the statement “This feature indicates that the inverse Edelstein mechanism offers highly efficient spin-charge conversion due to the spin-momentum locking property of the TI surface states”.

The larger noise at T = 2.3 K might be related to our Janis He-3 measurement system. The temperature control is very stable below 2.2 K, and the temperature variation becomes larger from 2.2 - 8 K.

5) *Figure 5 shows magnetization angle dependence of the voltage due to spin pumping. The magnetic field angle dependence is the same with conventional spin Hall effect [Ando et al, PRB 78 014413 (2008)]. It is difficult to claim that the voltage is caused by Edelstein effect from the magnetic field angle dependence.*

Reply: We thank the referee for pointing out this issue. The voltage due to spin pumping exhibits a linear relationship as the function of the magnetization angle. This is totally different from the model based on the spin pumping and inverse spin Hall effect (Ando et al, PRB 78 014413 (2008)) (blue line in the figure below). We have also tried to use the Cosine function to fit our results (red line in the figure below), but the Cosine function also fails. We believe that the complete understanding of the V_{SP} as a function of the θ_M needs further theoretical studies.

To make this clear, we have added a new section (S4. The magnetization angle dependence of the voltage due to spin pumping) and the figure above as Fig. S5 into the supplementary information

Response to Referee #3:

1) *The authors claim that the observed spin signal stems from the IEE associated with the spin-momentum locking of the topologically non-trivial surface states in SmB₆. However, the IEE can be observed in topologically "trivial" two dimensional states induced by strong Rashba splitting in the band structure. (For instance, see J.C. Sanchez et al., Nat. Commun. 4 2944 (2014).)*

Although extensive experimental studies support the presence of the surface states in SmB₆, the topologically non-trivial aspects have been confirmed only by a few reports, including the torque magnetometry (G. Li et al. Science 346, 1208 (2014)), spin resolved ARPES (N. Xu et al., Nat. Commun. 5, 4566 (2014)), and one-dimensional chiral magnetotransport (Y. Nakajima et al. Nat. Phys. 12, 213 (2016)). Besides, even polarity-driven topologically-trivial surface states has been observed by ARPES measurements (Z. Zhu, et al., Phys. Rev. Lett. 111, 216402 (2013)). It seems to me that the surface of SmB₆ is quite reactive, variously terminated and can reconstruct, providing different things observed by various groups. In this situation, it is unclear whether the spin-momentum locked surface states in SmB₆ give rise to the IEE.

However, it would be instructive to know if the observed spin signal quantitatively exclude the presence of topologically "trivial" surface states in SmB₆. Otherwise, the authors should use moderate wording to describe the origin of the spin signal.

Reply: We thank the referee for pointing out this issue. We agree with the referee that we cannot exclude the possible presence of the topologically "trivial" surface states in SmB₆ and we have revised our manuscript following the referee's suggestion using moderate wording. These revisions include:

- 1) In the abstract, we have changed that sentence on page 2, line 8 to be “*we experimentally demonstrate the spin injection and observe inverse Edelstein effect in the surface states of topological Kondo insulator, SmB₆.*”
- 2) In the introduction, we have changed the sentence on page 3, line 12 to be “*The temperature... spin voltage are consistent with the spin-momentum locking properties of the surface states, which have been shown to be topological in previous studies.*”
- 3) In the summary section, we have revised it to be: “*Our experimental results strongly support the demonstration of spin injection and the observation of the inverse Edelstein effect in the surface states of SmB₆. The temperature and magnetization angle dependence, as well as the sign of the spin-to-charge conversion are well consistent with spin-momentum locking properties of the surface states, which have been shown to be topological with the counter-clockwise spin textures for the electron bands in previous studies [Xu et al, Nat. Commun. 5, (2014) and Yu et al, arXiv:1603.09677 (2016)]. Since the detailed spin textures of the Rashba surface states have not been reported yet, it is premature to exclude any contribution from the Rashba-split surface states at the current stage. To fully understand this, further studies, including the*

detailed spin textures from spin-APERS measurements of the Rashba surface states and the quantitative theoretical calculation of the contributions from topological and Rashba surface states, are needed.”

2) As mention above, the Fermi surfaces observed by ARPES measurements seem to be inconsistent with each other. What literature is the schematic drawing in Fig.1b based on? Experiment or theory? The authors should cite the papers on which the drawing is based.

Reply: We thank the referee for pointing out this issue. We have revised the figure captions to be “**b**, *Schematic drawing of the spin-momentum locking properties of the topological surface states at the X and Γ points based on previous photoemission spectroscopy measurements and DFT calculations [Xu et al, Nat. Commun. 5, (2014) and Yu et al, arXiv:1603.09677 (2016)].”*

3) It would be instructive to compare the value of the observed spin signal with that observed in the spin injection into weakly correlated TIs. For instance, similar spin-charge conversion in BSTS/Ni81Fe19 device has been reported (Y. Shiomi et al., PRL 113, 196601 (2014)), showing much larger spin signal (~10 micro volts) induced by much smaller microwave power than the present work. This fact appears to evince that the IEE in SmB6 might be induced by trivial Rashba splitting surface states because one of the pair of splitting bands counteracts the effect of the other. The authors should elaborate on the difference between the signals observed in SmB6 and weakly correlated TI devices.

Also, the observed sign of the spin signal in SmB6 is opposite to that in the BSTS/Ni81Fe19 device. I'm curious to know if it's due to intrinsic mechanism or just the difference of notation.

Reply: We thank the referee for his/her suggestion. We agree that our signal is smaller than the study on BSTS (Shiomi et al., PRL 113, 196601 (2014)). There are several possible reasons that account for this. First, the actual microwave power depends on the distance between the coplanar waveguides and the Py electrode and the loss due to the RF cable and connectors from room temperature to below 2 K. In our manuscript, the power indicates the output power from the signal generator and we do not know the actual power that Py absorbed. Second, the spin pumping efficiency could be very different due to the interface details, such as some oxide layer or spin flip scattering at the interface.

We found that it is hard to claim whether the signal is due to topological surface states or Rashba-split states based only on the signal size. It is first predicted that inverse Edelstein effect in Rashba surface states is larger than the topological surface states (Hong, et al, Phys. Rev. B 86, 085131, (2012)). However, recent experimental results reporting larger spin signal for Rashba surface states on Bi₂Se₃ based topological insulator (Yang, et al, arXiv:1605.04149).

Besides, the observed sign of the spin signal in SmB₆ is opposite to that in the BSTS/Ni₈₁Fe₁₉ device, which is expected due to the opposite directions of the spin textures of SmB₆ and BSTS.

To make this clear, we have added the ground voltage reference in Fig. 1a and the following sentences on page 4, line 17: “*The V_{SP} exhibits a positive sign for positive magnetic field and the sign of the spin-to-charge conversion in the surface states of the SmB₆ is theoretically expected from the counter-clockwise spin textures for the electron band of the topological surface states [Shiomi, et al, PRL 113, 196601 (2014) and Rojas-Sánchez, et al, PRL, 116, 096602 (2016)]. The counter-clockwise spin textures have been both shown by spin-APERS measurements and DFT calculations [Xu et al, Nat. Commun. 5: 4566(2014) and Yu et al, arXiv:1603.09677 (2016)].*”

- 4) *The presence of the bulk carriers is a major obstacle to see the EE/IEE in (weakly correlated) TIs. Truly bulk-insulating at low temperatures, Kondo insulator SmB₆ is one of the suitable materials for the spin injection and detection through the IEE. Again, to compare SmB₆ with other TIs, it would be very helpful to quantify the spin-injection efficiency, which can be associated with leakage into bulk carriers, if possible.*

Reply: We thank the referee for pointing out this issue. We have done the comparison in the reply to the #3 issue earlier. For the spin injection efficiency, we have estimated the spin mixing conductance by comparing the Gilbert damping of Py on SmB₆ and SiO₂ substrate by simply neglecting film roughness and the interface spin-flip scattering contributions to the total Gilbert damping. The upper bound for the effective spin mixing conductance is about equal to $(1.0 \pm 0.1) \times 10^{20} \text{ m}^{-2}$. We have revised the supplementary information S3 accordingly.

- 5) *It is known that a native surface oxide layer is observed by X-ray photoelectron spectroscopy for SmB₆ (W.A. Phelan et al., Phys. Rev. X 4, 031012 (2014)), which could affect the efficiency of spin injection. How did the author treat the surface of SmB₆ while fabricating the device?*

Reply: We thank the referee for pointing out this issue. We did not polish the surface after we received the samples from our collaborator. The reason is that we were afraid to damage the surface states since we noted that in a previous report done by our colleagues (Li, et al, Science 346, 1208-1212, (2014)), they found “*polishing samples reduce the sample mobility by an order of magnitude to $\sim 100 \text{ cm}^2/\text{Vs}$ ”.* Hence, some native surface oxide layer might be on the surface of SmB₆. And this native surface oxide layer could strongly affect the spin pumping efficiency. We consider removing the native oxide layer on SmB₆ by polishing as the next study.

- 6) *It would be instructive to know the typical length scales in SmB₆, such as the sample thickness, the spin diffusion length in the surface state, and the penetration depth of the surface states, to figure out the properties of the device in the present*

work.

Reply: We thank the referee for pointing out this issue. The sample thickness is ~ 0.5 mm, we have added this into the methods section. Unfortunately, we are not able to determine the spin diffusion length of the surface states and the penetration depth of the surface states using our experimental setup at the current stage.

Detailed responses to Referees

Referees' comments in blue and italics and our responses in black.

Response to Referee #1:

Reviewer #1 (Remarks to the Author):

I very much appreciate the efforts of the authors to address the concerns I have raised in my report. Nevertheless, based on the author's response I believe that the current work only adds to the controversy surrounding SmB6 on the possible topological nature of the surface states as opposed to resolving it. Therefore, I do not recommend this paper for the publication in Nature Communications.

Reply: We thank the referee for his/her appreciation of our efforts.

Although we did not attempt to resolve the controversies about the nature of the surface states (topological vs. Rashba), our findings reveal unique properties (spin-momentum locking) of the surface states of topological Kondo insulator for the first time. We do not believe that our results by any means add confusion to the problem.

Our work reports the spin injection and the observation of inverse Edelstein effect of the spin-momentum locked surface states in topological Kondo insulator SmB₆. The novelty and importance of work has already been stated by both referee #2 and referee #3, “*the research topic in the current study is very important and timely for spintronics applications*” and “*This work is intriguing, the manuscript is well-written, and scientifically solid data clearly support the observation of spin injection and detection in the surface states of the strongly correlated material for the first time.*”

Overall, we believe that the novelty and importance of these results make our manuscript suitable to be published in Nature Communications.

Response to Referee #2:

Reviewer #2 (Remarks to the Author):

Authors have addressed the points raised in the previous review with varying degrees of satisfaction. I believe that the detected signals are caused by spin-to-charge conversion in surface state of topological Kondo Insulator SmB6. So I would be happy to see the manuscript accepted for publication at Nature Communications.

However, I could not understand whether the amplitude of detected signal is reasonable compared with ideal case. So authors should compare the amplitude of generated voltage (or conversion efficiency) with previous works in several topological

insulators like alpha-Sn (Phys. Rev. Lett. 116, 096602). If you found that detected signal is tiny, you should add the comments about the result in main text.

Reply: We thank the referee for his/her positive comments and recommendation of publication in Nature Communications.

We apologize that we did not address this minor point clearly enough in our previous response. To make it more clearly, we have added the following sentence on page 6, line 7. “At 0.8 K, when only spin-momentum locked surface states exist, the spin signal is ~ 42 nV. This value is relatively small compared to previous studies on $\text{Bi}_{1.5}\text{Sb}_{0.5}\text{Te}_{1.7}\text{Se}_{1.3}$ and $\alpha\text{-Sn}$ [Phys. Rev. Lett. 116, 096602 (2016) and PRL 113, 196601 (2014)], which could be related to the spin pumping efficiency and/or the spin-to-charge conversion efficiency and needs further studies (Supplementary Note 3).”

Response to Referee #3:

Reviewer #3 (Remarks to the Author):

The authors have provided a reasonable and satisfactory reply to my comments, and the manuscript has been improved. Therefore, the paper meets all the standards for publication in Nature Communications, and I recommend publication as it is.

Reply: We thank the referee for his/her positive comments and recommendation of publication in Nature Communications as it is!